# MAFLD and CKD: An Updated Narrative Review

**DOI:** 10.3390/ijms23137007

**Published:** 2022-06-23

**Authors:** Alessandro Mantovani, Rosa Lombardi, Filippo Cattazzo, Chiara Zusi, Davide Cappelli, Andrea Dalbeni

**Affiliations:** 1Section of Endocrinology, Diabetes and Metabolism, Department of Medicine, University and Azienda Ospedaliera Universitaria Integrata of Verona, 37126 Verona, Italy; chiara.zusi@gmail.com (C.Z.); davide.cappelli3@gmail.com (D.C.); 2Unit of Internal Medicine and Metabolic Disease, Fondazione IRCCS Ca’ Granda Ospedale Maggiore Policlinico, 20122 Milan, Italy; rosa.lombardi@unimi.it; 3Department of Pathophysiology and Transplantation, Università degli Studi di Milano, 20133 Milan, Italy; 4Section of General Medicine C and Liver Unit, University and Azienda Ospedaliera Universitaria Integrata of Verona, 37126 Verona, Italy; cattazzo.f@gmail.com (F.C.); andrea.dalbeni@aovr.veneto.it (A.D.); 5Pediatric Diabetes and Metabolic Disorders Unit, Department of Surgical Sciences, Dentistry, Pediatrics and Gynaecology, University Hospital of Verona, 37100 Verona, Italy

**Keywords:** non-alcoholic fatty liver disease, NAFLD, non-alcoholic steatohepatitis, NASH, metabolic associated fatty liver disease, MAFLD, chronic kidney disease, CKD

## Abstract

Accumulating evidence now indicates that non-alcoholic fatty liver disease (NAFLD), which is the most common chronic liver disease observed in clinical practice worldwide, is independently associated with an increased risk of incident chronic kidney disease (CKD). Given that NAFLD is linked to insulin resistance, obesity and type 2 diabetes mellitus, an international panel of experts have recently proposed a name change from NAFLD to metabolic associated fatty liver disease (MAFLD). Since the diagnostic criteria for NAFLD and MAFLD are different, observational studies assessing the potential concordance (or even superiority) of MAFLD, compared with NAFLD, in detecting patients at increased risk of hepatic and extra-hepatic complications (including CKD) are required. Hence, in the last two years, some observational studies have investigated the potential relationship between MAFLD and CKD. The result is that, at present, evidence regarding the concordance or even superiority of MAFLD, compared with NAFLD, in detecting patients at higher risk of CKD is still preliminary, although some data indicate that MAFLD identifies patients with CKD as accurately as NAFLD. In this narrative review, we will discuss: (a) the epidemiological evidence assessing the association between NAFLD and risk of incident CKD, (b) the epidemiological data investigating the association between MAFLD and risk of CKD and (c) the biological mechanisms underlying the association between NAFLD/MAFLD and CKD.

## 1. Introduction

Non-alcoholic fatty liver disease (NAFLD) includes various progressive pathological conditions, such as simple steatosis, non-alcoholic steatohepatitis (NASH), advanced fibrosis and cirrhosis, in patients without excessive alcohol consumption and secondary causes of chronic liver disease [1]. Currently, NAFLD is the most frequent chronic liver disease seen in clinical practice worldwide [1]. In this setting, several epidemiological studies and some meta-analyses have estimated that NAFLD affects approximately 25–30% of adults in the general population [2,3], up to 70% of patients with type 2 diabetes mellitus (T2DM) [4] and almost all patients with moderate/severe obesity [5]. NAFLD is closely linked to insulin resistance, adiposity and T2DM leading to the development of adverse hepatic and extra-hepatic outcomes [1]. In this context, it has become evident that NAFLD is a “multisystem disease” [6], which is associated with hepatic dysfunction or hepatocellular carcinoma (HCC) [1], but also with an increased risk of developing cardiovascular disease (the main cause of death in these patients) [7], T2DM [8] and chronic kidney disease (CKD) [9].

Based on this essential background, in the past two years, many Experts in the field and some scientific Societies, although not all [10,11,12], have proposed to change the terminology, switching from NAFLD to metabolic associated fatty liver disease (MAFLD) [13,14]. Accordingly, the diagnosis of MAFLD can be performed based on the presence of hepatic steatosis (detected by serum biomarker scores, imaging techniques or liver biopsy) and at least one of the following metabolic criteria: (a) overweight/obesity, (b) T2DM, and (c) metabolic dysregulation, i.e., at least two additional factors amongst increased waist circumference, hypertension, hypertriglyceridemia, low serum HDL-cholesterol levels, impaired fasting glucose, insulin resistance or subclinical inflammation [13,14]. Interestingly, some observational studies have recently reported that the definition of MAFLD, compared with NAFLD, improves the identification of patients at higher risk of developing hepatic and extra-hepatic complications [15,16,17], including CKD.

CKD is a long-term condition, frequently observed in clinical practice, which is associated with an increased risk of morbidity and mortality, as well as with a high economic cost [18]. The global estimated prevalence of CKD is approximately 10% [18,19,20,21], resulting in 1.2 million deaths and 28 million years of life lost each year [18]. Alarmingly, by 2040 CKD might become the fifth leading cause of death worldwide [18]. As consequence, the identification of novel and additional factors, associated with the development and progression of CKD, is relevant to find strategies that might reduce the clinical impact of CKD [21]. In this setting, the validation of NAFLD/MAFLD as an independent risk factor of CKD is important [21]. Hence, since the diagnostic criteria for NAFLD and MAFLD are different [13,14], some observational studies assessing the concordance (or even superiority) of MAFLD in the identification of patients at higher risk of CKD, compared with NAFLD, have been published in the last years.

In this narrative review, we will discuss (a) the epidemiological evidence assessing the association between NAFLD and risk of incident CKD, (b) the epidemiological studies investigating the association between MAFLD and risk of CKD and (c) the biological mechanisms underlying the association between NAFLD/MAFLD and risk of CKD.

## 2. Association between NAFLD and Risk of Incident CKD

Over the last decades, several epidemiological studies and some meta-analyses have clearly reported that NAFLD (detected by blood biomarkers/scores, imaging techniques, International Classification of Diseases codes or liver biopsy) is associated with an increased risk of incident CKD, independent of established CKD risk factors, diabetes-related variables and other potential confounders [21,22,23,24,25]. Notably, as previously mentioned, the association between NAFLD and CKD have relevant clinical implications, as both NAFLD and CKD are two important global health problems with the worryingly direction to become more and more frequent worldwide [21]. As reported in Table 1, to date there are (at least) three meta-analyses investigating the association between NAFLD and risk of incident CKD [9,26,27]. In the 2014 meta-analysis including 33 eligible cross-sectional and longitudinal studies, Musso et al. reported that NAFLD (detected by liver enzymes, imaging techniques or liver biopsy) was independently associated with an increased risk of prevalent (random effects odds ratio [OR] 2.12, 95% confidence interval 1.69–2.66) and incident (random effects hazard ratio [HR] 1.79, 95% confidence interval 1.65–1.95) CKD [26]. In addition, in that study, the authors also found that patients with NASH had a higher risk of both prevalent (random effects OR 2.53, 95% confidence interval 1.58–4.05) and incident (random effects HR 2.12, 95% confidence interval 1.42–3.17) CKD, compared with those with simple steatosis [26]. Importantly, patients with advanced fibrosis had the highest risk of both prevalent (random effects OR 5.20, 95% confidence interval 3.14–8.61) and incident (random effects HR 3.29, 95% confidence interval 2.30–4.71) CKD [26]. In the 2018 meta-analysis including nine longitudinal studies, Mantovani et al. documented that NAFLD patients (as detected by imaging techniques) had a higher risk of incident CKD, compared with those without NAFLD, over a median follow-up of nearly 5 years (random-effects HR 1.37, 95% confidence interval 1.20–1.50) [27]. Again, in the meta-analysis by Mantovani et al., patients with advanced forms of NAFLD (detected by ultrasound and/or non-invasive fibrosis markers) had the highest risk of incident CKD (random-effects hazard ratio 1.50, 95% confidence interval 1.25–1.74) [27]. In the 2022 meta-analysis involving 13 longitudinal studies for a total of 1,222,032 individuals (28.1% with NAFLD as detected by blood biomarkers/scores, International Classification of Diseases codes, imaging techniques or biopsy), the same research group additionally confirmed that NAFLD was significantly associated with an increased risk of incident CKD (random-effects HR 1.43, 95% confidence interval 1.33–1.54), over a median follow-up of nearly 10 years [9].

As reported in Table 1, the three aforementioned meta-analyses display some differences in the overall effect of NAFLD on the risk of CKD. In particular, the effect of NAFLD on the risk of incident CKD seems to be higher in the meta-analysis by Musso et al. [26]. We believe that these differences might be due, at least in part, to some specific factors, such as inclusion/exclusion criteria, sample size, duration of follow-up, the definition of incident CKD cases, as well as by techniques used to detect hepatic steatosis.

That said, overall, the evidence available so far clearly shows that NAFLD is independently associated with an increased risk of incident CKD in patients with and without T2DM and that such risk increases in relation to the severity of the liver disease.

## 3. Association between MAFLD and Risk of CKD

As previously described, a new definition for NAFLD has been recently proposed, namely MAFLD. Supported by the known association between NAFLD and CKD [21,28], data regarding the relationship between MAFLD and renal impairment are relevant [23]. Indeed, hepatic fat is a risk factor for renal impairment. Hypertension, obesity, T2DM, but also viral hepatitis, are associated with CKD [29,30,31,32]. Alcohol consumption shows a U-shaped association with CKD, as moderate drinkers seem to have a lower CKD prevalence compared with non-drinkers and heavy drinkers [33,34,35]. In addition, some evidence also reports that non-drinkers and those who drink daily could have a higher risk of CKD compared with weekly drinkers [35]. Hence, given that alcohol and dysmetabolism are included in the definition of MAFLD [36], the relationship between MAFLD and the risk of CKD has been assessed [23] (Table 2).

In a Korean study 268,946 participants attending the National Health Insurance Service health examinations between 2009 and 2015 were prospectively followed-up for nearly 5 years. Despite the similar prevalence of NAFLD and MAFLD at baseline (about 30%), MAFLD subjects had a higher risk (HR 1.18) of developing CKD (diagnosed as estimated glomerular filtration rate (eGFR) < 60 mL/min/1.73 m^2^ and/or proteinuria) compared with those with NAFLD [37]. Notably, this risk further increased (HR 1.36) when NAFLD and MAFLD coexisted [37]. The superiority of MAFLD, compared with NAFLD, in identifying patients at higher risk of CKD was also confirmed by another observational study of 12,571 individuals from the Third National Health and Nutrition Examination Survey (1988–1994) [38]. Again, despite similar prevalences of MAFLD and NAFLD (30.2% and 36.2%, respectively), MAFLD individuals had lower eGFR values (74.9 ± 18.2 vs. 76.5 ± 18.2 mL/min/1.73 m^2^, respectively, *p* < 0.001), as well as a higher prevalence of CKD (29.6% vs. 26.6%, respectively, *p* < 0.05), compared with NAFLD ones [38]. Interestingly, the relationship between MAFLD and CKD was significant after adjustment for multiple CKD risk factors, such as age, gender, ethnicity, alcohol intake and presence of T2DM (adjusted-OR 1.12, 95% confidence interval 1.01–1.24) [38]. Conversely, NAFLD was not independently associated with an increased risk of prevalent CKD (adjusted-OR 1.06, 95% confidence interval 0.96–1.17) [38]. Similar evidence was also reported by Hashimoto et al. in an observational study of 27,371 participants, where MAFLD, but not NAFLD, was independently associated with both prevalent (OR 1.83; 95% confidence interval 1.66–2.01) and incident CKD (HR 1.24, 95% confidence interval 1.14–1.36), over a mean follow-up of 4.6 years [39]. Hashimoto et al. speculated that these findings might be partly explained by insulin resistance [39]. However, given that insulin resistance is also associated with NAFLD, we believe that the study by Hashimoto et al. [39] did not clarify the lack of an association between NAFLD and renal impairment.

Some observational studies did not show a significant association between MAFLD and CKD. For instance, in a cohort study enrolling 4869 US individuals (21% with CKD) from the National Health and Nutrition Examination Survey (NHANES) III (2017–2018), Deng et al. reported that MAFLD (assessed by liver ultrasound transient elastography) was not independently associated with CKD [40]. In that study, independent predictors of CKD were T2DM, hypertension and hyperuricemia, thus suggesting that the potential link between MAFLD and CKD may be partly attenuated by specific metabolic abnormalities [40]. In addition, it is also possible to speculate that in the study by Deng et al. [40] the diagnosis of hepatic steatosis by liver ultrasound transient elastography might have created a selection bias among MAFLD patients, thus explaining the lack of a significant association.

Zhang et al. evaluated the prevalence of MAFLD and NAFLD in 19,617 non-pregnant adults aged ≥20 years from the cross-sectional NHANES database, focusing on four different periods: 1999 to 2002, 2003 to 2006, 2007 to 2010, and 2011 to 2016 [41]. The authors reported an increasing prevalence of MAFLD (ranging from 28% to 36%) and NAFLD (ranging from 26% to 33%) over time. Interestingly, the prevalence of CKD, defined as eGFR <60 mL/min/1.73 m^2^ and urinary albumin-to- creatinine ratio (ACR) ≥30 mg/g, increased similarly in patients with MAFLD (from 17.9% to 18.7%) and in those with NAFLD (from 18.2% to 18.8%) over the four periods [41]. However, the risk of having CKD in the MAFLD group was only moderately higher than in the NAFLD group (OR 1.67 vs. 1.59, respectively) [41]. Finally, in a community-based cohort study involving 6873 Chinese individuals from Shanghai Nicheng Cohort, who were followed up for nearly 5 years, Liang et al. showed a similar prevalence of NAFLD and MAFLD (∼40%) at baseline, as well as a similar risk of incident CKD in patients with MAFLD (risk ratio 1.71, 95% confidence interval 1.44–2.04) and in those with NAFLD (risk ratio 1.70, 95% confidence interval 1.43–2.01) [42]. However, it is important to note that in the study by Liang et al. approximately 5% of participants had MAFLD but also an excessive alcohol consumption (i.e., >140 g/week for men and >70 g/week for women) [42]. Hence, given the potential effects of excessive alcohol consumption on fatty liver [36], CKD [35] and metabolic dysfunction [43], we believe that an accurate definition of alcohol consumption is relevant to examining the differences between MAFLD and NAFLD, including for the risk of CKD.

## 4. Is MAFLD Concordant (or Superior) to NAFLD in Detecting Patients at Higher Risk of CKD?

At present, studies regarding the concordance or even superiority of MAFLD, compared with NAFLD, in detecting patients at higher risk of renal impairment indicate that MAFLD identifies patients with CKD as accurately as NAFLD. However, some important aspects need to be mentioned about this topic. First, the evidence available so far might be partly invalidated by the fact that hepatic steatosis was detected by non-invasive markers (such as fatty liver index) in some studies. Second, the results on this topic mainly refer to the NHANES program and, hence, observational studies involving other patient populations are warranted. Third, while there are data regarding the role of NAFLD on the progression of CKD [44], to date information about the role of MAFLD on the worsening of kidney function over time is scarce. In addition, although accumulating evidence also indicates that in NAFLD patients the improvement in liver histology (especially by lifestyle modifications) is associated with amelioration in kidney function [45], there is no information about this issue in MAFLD patients. Fourth, since obesity, T2DM, and metabolic dysfunction are included in the MAFLD definition [36], it is difficult to determine exactly the specific role of liver disease (from hepatic steatosis to steatohepatitis to advanced fibrosis) on the development and progression of CKD. Again, the presence of viral hepatitis that is associated with CKD can coexist with MAFLD definition [23]. As consequence, they might potentially modulate the association between MAFLD and CKD. Fifth, but not for importance, one should consider the impact of alcohol on fatty liver disease [36], but also on CKD [35]. To date, indeed, there is no robust evidence regarding a safe threshold for alcohol in individuals with liver disease [36,46]. Conversely, for CKD, some evidence indicates a U-shaped association between alcohol and CKD [35]. Therefore, we believe that the quantification and frequency of alcohol consumption is essential for assessing the relationship between MAFLD and CKD. Finally, the use of non-invasive markers of liver fibrosis (such as fibrosis-4 [FIB-4] score, NAFLD fibrosis score, aspartate aminotransferase to platelet ratio index [APRI], or AST-to-ALT ratio) in MAFLD patients has not yet been validated [15]. As consequence, the non-invasive markers of liver fibrosis might not be useful in the evaluation of the association between severe forms of MAFLD and CKD.

Doubtless, our narrative review highlights the need for further studies, possibly prospective, to clarify better the association between MAFLD and CKD progression.

## 5. Putative Mechanisms Underpinning the Association between NAFLD/MAFLD and Risk of CKD

In literature, several narrative reviews have already described the biological mechanisms underpinning the association between NAFLD/MAFLD and the risk of CKD [21,23,24,25]. Here, we will limit ourselves to describing the salient aspects of this topic (Figure 1). Growing experimental and clinical evidence indicates that NAFLD/MAFLD and its advanced forms exacerbate (a) the systemic insulin resistance (via the secretion of hepatokines, such as fibroblast growth factor-21), (b) the atherogenic dyslipidaemia and hypertension, (c) the activation of the renin-angiontensin system (RAS) which, in turn, is associated with endothelial dysfunction, and (d) the release, into the bloodstream, of several mediators able to promote a chronic pro-inflammatory and pro-coagulant state [21,23,24,25]. All these factors may play a role in the pathophysiology of CKD [21,23,24,25]. In addition, interestingly, preliminary studies also indicate that some specific genetic polymorphisms, especially rs738409 C>G p.I148M in the *PNPLA3* (patatin-like phospholipase domain-containing protein 3) gene, have an important role in the development and progression of NAFLD [47], MAFLD [48], but also a potential role in the development of kidney abnormalities [23,49]. Other potential genetic polymorphisms associated with NAFLD and potentially linked to kidney abnormalities are described in detail in the 2022 review by Wang et al. [23]. A better comprehension of these complex biological mechanisms may lead to novel targets for the treatment of NAFLD/MAFLD but also of CKD [21,23,24,25].

## 6. Conclusions

The use of the new definition MAFLD, compared to NAFLD, offers numerous advantages in clinical, epidemiological, and research terms [15,16,50]. First, the diagnosis of MAFLD is inclusive, identifies a more “homogeneous” patient group, and emphasizes the role of metabolic dysfunction in the pathogenesis of this condition. Second, preliminary data show that MAFLD criteria may better identify adults with liver steatosis at higher risk of hepatic and extra-hepatic complications [15,51]. Third, preliminary evidence suggests that the MAFLD definition might better capture patients who might benefit from an evaluation of genetic risks for fatty liver, as *PNPLA3* rs738409 seems to be associated with MAFLD [48], but also with the development of kidney abnormalities [23,49].

To date, there is evidence (mainly provided by cross-sectional studies) suggesting that MAFLD identifies patients at higher risk of CKD as accurately as NAFLD. However, it is important to note that some studies have confuted this postulation (Table 2). Hence, we believe that larger and prospective studies are timely needed to establish the association between MAFLD and CKD risk. In addition, studies involving European and Asian individuals, but also enrolling specific patient groups (such as elderly patients or lean ones), are required to get further information about the consistency of the association between MAFLD and risk of CKD. That said, however, regardless of the definition of MAFLD or NAFLD, the presence of hepatic fat seems to identify patients at higher risk of CKD [23]. Consequently, a multidisciplinary approach for managing and treating NAFLD/MAFLD patients is relevant, especially to prevent the development of hepatic and extra-hepatic complications [23,52,53]. At present, no specific curative treatment is available for NAFLD/MAFLD and CKD [21,23]. However, although lifestyle modifications can be difficult to maintain over time, hypocaloric diet and exercise can induce weight loss which, in turn, promotes the regression of liver disease, as well as the reduction of the risk of CKD, CVD, and other metabolic comorbidities [21,54]. In addition, there is accumulating scientific interest in the potential role of some glucose-lowering agents (such as pioglitazone, glucagon-like peptide-1 receptor agonists, and sodium-glucose cotransporter-2 inhibitors) in patients with NAFLD/MAFLD, as they exert benefits on hepatic fat content and steatohepatitis, as well as benefits on cardiorenal outcomes independent of the presence of T2DM [21,23,54,55]. Other agents are being studied with the potential to benefit both CKD and NAFLD/MAFLD [21,23,54,55]. Meanwhile, we should be aware of the possibility of CKD in patients with NAFLD/MAFLD. As consequence, the assessment of renal function over time is mandatory in these patients [21].

## Figures and Tables

**Figure 1 ijms-23-07007-f001:**
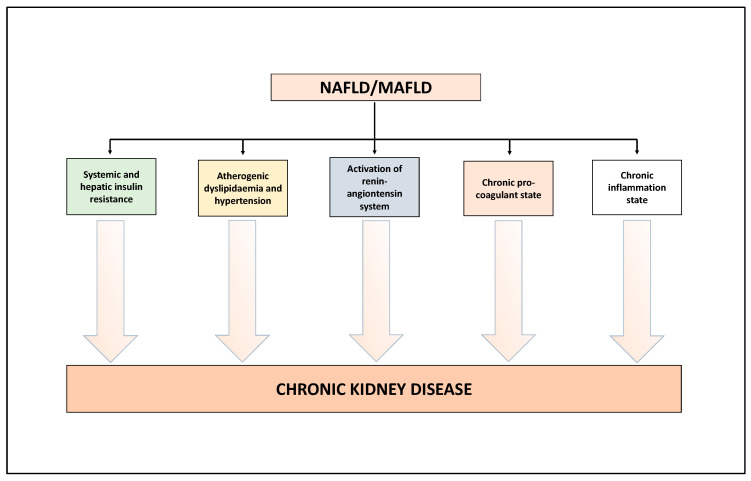
Putative biological mechanisms underlying the association between NAFLD/MAFLD and risk of CKD. See text for details.

**Table 1 ijms-23-07007-t001:** Main systematic reviews and meta-analyses assessing the association between non-alcoholic fatty liver disease (NAFLD) and risk of incident chronic kidney disease (CKD).

Reference	Study Characteristics	Definition of NAFLD	Definition of CKD	Main Results
[26]	Systematic review and meta-analyses of 33 observational studies (20 cross-sectional ones and 13 longitudinal ones) for a total of 63,902 individuals	Liver enzymes, ultrasonography and liver biopsy	One or more of the following criteria: eGFR < 60 mL/min/1.73 m^2^ ACR ≥ 30 mg/g Morning dipstick proteinuria ≥1	NAFLD was associated with a higher risk of prevalent (random effects OR 2.12, 95% CI 1.69-2.66) and incident (random effects HR 1.79, 95% CI 1.65–1.95) CKD NASH was associated with a higher risk of prevalent (random effects OR 2.53, 95% CI 1.58–4.05) and incident (random effects HR 2.12, 95% CI 1.42–3.17) CKD than simple steatosis Advanced fibrosis was associated with a higher prevalent (random effects OR 5.20, 95% CI 3.14–8.61) and incident (random effects HR 3.29, 95% CI 2.30–4.71) CKD than non-advanced fibrosis
[27]	Systematic review and meta-analyses of nine longitudinal studies for a total of 96,595 individuals (34.1% with NAFLD). Median follow-up: 5.2 years	Liver enzymes, fatty liver index and ultrasonography	eGFR < 60 mL/min/1.73 m^2^ and/or overt proteinuria	NAFLD was associated with a higher risk of incident CKD (random effects HR 1.37, 95% CI 1.20–1.53) More severe NAFLD (defined as a high-intermediate NAFLD fibrosis score or elevated serum GGT levels among individuals with ultrasound-diagnosed NAFLD) was associated with a higher risk of incident CKD (random effects HR 1.50, 95% CI 1.25–1.74)
[9]	Systematic review and meta-analyses of 13 longitudinal studies (with a follow-up duration of ≥1 year) for a total of 1,222,032 individuals (28.1% with NAFLD). Median follow-up: 9.7 years	Liver enzymes, fatty liver index, imaging techniques, ICD-9 codes, and liver biopsy	eGFR < 60 mL/min/1.73 m^2^ and/or overt proteinuria CKD stage ≥ 3 identified by the ICD-9 codes	NAFLD was associated with a higher risk of incident CKD (random effects HR 1.43, 95% CI 1.33 to 1.54)

Abbreviations: ACR, albumin-to-creatinine ratio; CI, confidence interval; CKD, chronic kidney disease; eGFR, estimated glomerular filtration rate; FLI, fatty liver index, HR, hazard ratio; ICD, International Classification of Diseases; NAFLD, non-alcoholic fatty liver disease; NASH, non-alcoholic steatohepatitis; OR odds ratio.

**Table 2 ijms-23-07007-t002:** Main observational studies evaluating the association of MAFLD and NAFLD with chronic kidney disease (CKD).

Reference	Study Characteristics	Definition of NAFLD/MAFLD	Prevalence of NAFLD and MAFLD	Definition of CKD	Main Results
[37]	Cross-sectional and prospective (mean follow-up 5.1 years) study: 268,946 US participants attending the National Health Insurance Service health (2009–2015) in the USA	Fatty liver index	NAFLD: 27% MAFLD: 33%	eGFR < 60 mL/min/1.73 m^2^ and/or proteinuria (i.e., ≥trace on dipstick test)	Patients with MAFLD had a significantly higher risk of developing CKD (adjusted HR 1.64, 95% CI 1.44–1.88) than patients with NAFLD. This relationship was maintained after adjustments for confounding factors (adjusted HR 1.18, 95% CI 1.01–1.39). The risk of incident CKD was even higher in those with overlapping fatty liver disease
[38]	Cross-sectional study: 12,571 US individuals included in the Third National Health and Nutrition Examination Survey (1988–1994) in the USA	Ultrasonography	NAFLD: 36% MAFLD: 30%	eGFR < 90 mL/min/1.73 m^2^ and or urinary albumin-to-creatinine ratio (ACR) ≥3 mg/mmol	MAFLD individuals had lower eGFR values (74.96 ± 18.21 vs. 76.46 ± 18.24 mL/min/1.73 m^2^, *p* < 0.001) and a greater prevalence of CKD (29.6% vs. 26.6%, *p* < 0.05) when compared to NAFLD individuals MAFLD was independently associated with an increased risk of CKD (OR 1.12, 95% CI 1.01–1.24), especially in the presence of advanced fibrosis as assessed by non-invasive markers (OR 1.34, 95% CI 1.06–1.69). NAFLD was not independently associated with an increased risk of CKD (OR 1.06, 95% CI 0.96–1.17).
[39]	Cross-sectional, prospective (median follow-up 4.6 years) study: 27,371 Japanese participants in medical health checkup program in Kyoto (2004–2014)	Ultrasonography	NAFLD: 2.3% MAFLD: 20.8%	eGFR < 60 mL/min/1.73 m^2^ and/or proteinuria	Compared to those without steatosis, patients with MAFLD had a higher risk of CKD (adjusted OR 1.83, 95% CI 1.66–2.01), whereas patients with NAFLD did not (adjusted OR 1.02, 95% CI 0.79–1.33) MAFLD was independently associated with an increased risk of incident CKD (adjusted HR 1.30, 95% CI 1.14–1.36), while NAFLD was not (adjusted HR 1.11, 95% CI 0.85–1.41)
[40]	Cross-sectional and prospective (median follow-up 4.6 years) study: 4869 US subjects from the National Health and Nutrition Examination Surveys (NHANES 2017–2018) in the USA	CAP >240 dB/min	MAFLD: 57%	eGFR < 60 mL/min/1.73 m^2^ and/or proteinuria	There was a higher prevalence of CKD in MAFLD subjects than in non-MALFD subjects (22.2% vs. 19.1%, respectively, *p* = 0.048). After 1:1 propensity score matching by gender, age, and race, MAFLD was not independently associated with CKD
[41]	Cross-sectional study: 19,617 US subjects from the National Health and Nutrition Examination Surveys in the USA over four periods: 1999–2002; 2003–2006; 2007–2010; 2011–2016	Fatty liver index >30	NAFLD 1999–2002: 26% 2003–2006: 29% 2007–2010: 32% 2011–2016: 33% MAFLD 1999–2002: 28% 2003–2006: 31% 2007–2010: 34% 2011–2016: 36%	eGFR < 60 mL/min/1.73 m^2^ and/or albumin-to-creatinine ratio (ACR) ≥30 mg/g	The risk of having CKD in the MAFLD group was only moderately higher than in the NAFLD group
[42]	Cohort study (median follow-up 4.6 years): 6873 Chinese subjects from The Shanghai Nicheng Cohort Study	Ultrasonography	NAFLD: 40% MAFLD: 46.7%	eGFR < 60 mL/min/1.73 m^2^ and/or albumin-to-creatinine ratio (ACR) ≥30 mg/g	Similar risks of incident CKD in the MAFLD group (relative risk 1.71, 95% CI 1.44–2.04) and NAFLD group (relative risk 1.70, 95% CI 1.43–2.01)

Abbreviations: ACR, albumin-to-creatinine ratio; CAP, controlled attenuation parameter; CI, confidence interval; CKD, chronic kidney disease; eGFR, estimated glomerular filtration rate; HR, hazard ratio; MAFLD, metabolic associated fatty liver disease; NAFLD, non-alcoholic fatty liver disease; OR, odds ratio.

## Data Availability

Not applicable.

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
