# Peer review of "MAFLD and CKD: An Updated Narrative Review"

_ijms, 2022, doi:10.3390/ijms23137007_

Round 1

Reviewer 1 Report

Overall, this narrative review is well done. It is engaging for the readers and the data are up-to-date. However, it needs to be corrected with respect to the following points.

MAFLD includes at-risk drinkers as already reported (PMID: 32930521). Reference number 40 (PMID: 34508601) misstates this definition; 4.7% of patients in Figure 2 fall into the MAFLD category. Thus, without an accurate definition of alcohol consumption, the difference between MAFLD and NAFLD cannot be examined. Furthermore, there are reports that daily alcohol consumption is involved in CKD (PMID: 35304050), so please cite the paper and discuss it as one of the future perspectives.

Author Response

Overall, this narrative review is well done. It is engaging for the readers and the data are up-to-date.

Authors’ response: Thank you very much for this supportive comment

MAFLD includes at-risk drinkers as already reported (PMID: 32930521). Reference number 40 (PMID: 34508601) misstates this definition; 4.7% of patients in Figure 2 fall into the MAFLD category. Thus, without an accurate definition of alcohol consumption, the difference between MAFLD and NAFLD cannot be examined. Furthermore, there are reports that daily alcohol consumption is involved in CKD (PMID: 35304050), so please cite the paper and discuss it as one of the future perspectives.

Authors’ response: Thank you very much for this comment. We have cited these manuscripts, as kindly suggested, as well as we have discussed these issues in the revised version of manuscript.

Reviewer 2 Report

This manuscript is a review paper that summarizes the new concept of association based on epidemiologic and research data on the correlation between MAFLD/NAFLD and CKD, and it is meaningful to share important information with related researchers.

However, there is a difference in the key correlation indicators between MAFLD/NAFLD and CKD, so there is some limitation in establishing a clear cross-correlation.

In addition, the following aspects need to be considered.

- Correction of repetitive sentences and more concise conclusions are required.

- The country of the first data in Table 2 should be changed from USA to Korea

- It is required to check the references in the text.

Author Response

This manuscript is a review paper that summarizes the new concept of association based on epidemiologic and research data on the correlation between MAFLD/NAFLD and CKD, and it is meaningful to share important information with related researchers.

Authors’ response: Thank you very much for this comment.

Correction of repetitive sentences and more concise conclusions are required.

Authors’ response: Thank you. As requested, in the revised manuscript, we have changed the repetitive sentences and shortened the conclusions.

The country of the first data in Table 2 should be changed from USA to Korea

Authors’ response: Thank you. We apologize for that. Now we have preferred to remove the column relative to country and add information about nationality in the column about study characteristics. 

It is required to check the references in the text.

Authors’ response: Thank you. We have done accordingly in the revised manuscript.

Reviewer 3 Report

Major

There is an interesting paper related to very important health problem.

However, the manuscript must be carefully checked and corrected by a native speaker familiar with medical/biological sciences.

Minor

“the impact of alcohol on kidneys damage is controversial” – it should be described in more detail as it seems to be related to the dose and time of high dose alcohol consumption

“MAFLD definition might better capture patients who might benefit  from an evaluation of genetic risks for fatty liver” –  What is benefit for the patient  related to evaluation of the genetic risk?   How  is it related to CKD risk “

“studies …..enrolling specific patient groups, are also required to get further information about the consistency of the association between  MAFLD and risk of CKD” – What kind “specific patients groups” do you have on mind ?

Author Response

However, the manuscript must be carefully checked and corrected by a native speaker familiar with medical/biological sciences.

Authors’ response: Thank you very much. The text has been corrected by a native speaker as requested.

Minor

“the impact of alcohol on kidneys damage is controversial” – it should be described in more detail as it seems to be related to the dose and time of high dose alcohol consumption

Authors’ response: Thank you very much for this suggestion. We have modified this section accordingly, as also suggested by Reviewer#1.  

“MAFLD definition might better capture patients who might benefit  from an evaluation of genetic risks for fatty liver” –  What is benefit for the patient  related to evaluation of the genetic risk?   How  is it related to CKD risk “.

Authors’ response: Thank you very much for this comment. Since PNPLA3 rs738409 seems to be associated with the development of MAFLD [J Hepatol 2021, 74, 974-977], in addition to NAFLD [J Hepatol 2018, 68, 268-279] and kidney abnormalities [Liver Int 2020, 40, 1130-1141], the evaluation of the genetic risk in MAFLD patients might help to identify patients at higher risk of developing hepatic complications, but also extrahepatic complications such as CKD (please see page 11 of revised manuscript).

“studies …..enrolling specific patient groups, are also required to get further information about the consistency of the association between  MAFLD and risk of CKD” – What kind “specific patients groups” do you have on mind ?

Authors’ response: Thank you very much for this comment. We have on mind elderly patients or lean patients. We have better specified this in the revised manuscript (please see page 12 of the revised manuscript).

Round 2

Reviewer 1 Report

Thank you for the correction. Now this paper deserves to be accepted for publication in International Journal of Molecular Sciences.

Author Response

Thank you very much for your supportive comment.